# Defining Culinary Medicine: A Call for Consensus on Competencies to Improve Nutrition

**DOI:** 10.3390/nu17091403

**Published:** 2025-04-22

**Authors:** Rani Polak, Beth Frates, Jacob Mirsky, Jennifer Trilk, Nathan Wood, Margaret Moore, Olivia Thomas, Edward M. Phillips

**Affiliations:** 1CHEF Coaching Program, Spaulding Rehabilitation Hospital, Department of Physical Medicine and Rehabilitation, Harvard Medical School, Boston, MA 02129, USA; 2Center of Lifestyle Medicine, Sheba Medical Center, Tel Hashomer, Tel Aviv 6997801, Israel; 3Spaulding Rehabilitation Hospital, Department of Physical Medicine and Rehabilitation, Harvard Medical School, Boston, MA 02129, USA; bethfrates@gmail.com (B.F.); ephillips1@mgh.harvard.edu (E.M.P.); 4Division of General Internal Medicine, Massachusetts General Hospital, Harvard Medical School, Boston, MA 02130, USA; jmirsky@mgh.harvard.edu; 5University of South Carolina School of Medicine Greenville, Greenville, SC 29605, USA; trilk@greenvillemed.sc.edu; 6Department of Internal Medicine, Yale School of Medicine, New Haven, CT 06510, USA; nathan.wood@yale.edu; 7Wellcoaches Corporation, Wellesley, MA 02482, USA; margaret@wellcoaches.com; 8Institute of Coaching, McLean Hospital, Belmont, MA 02478, USA; 9Nutrition Innovation and Implementation, Boston Medical Center, Boston, MA 02118, USA; olivia.thomas@bmc.org; 10VA Boston Healthcare System, Boston, MA 02130, USA

**Keywords:** Culinary Medicine, nutrition, health equity, climate change, ultra-processed food

## Abstract

Most premature adult deaths and chronic diseases, with their associated costs, are directly related to unhealthy behaviors, particularly poor nutrition. The 2022 White House Conference on Hunger, Nutrition, and Health emphasized the importance of nutrition equity and security as a key to preventing chronic diseases. What and how we eat also have important environmental impacts, with 26% of anthropogenic greenhouse gas emissions attributed to the total food supply chain, primarily ultra-processed food (UPF) production. A new paradigm is needed to better educate patients and the public to adopt healthier eating behaviors. Culinary education, emphasizing skills such as shopping, food storage, and meal preparation is a burgeoning field, aimed at reducing UPF consumption and improving nutrition while addressing cultural and socioeconomic factors. The term Culinary Medicine (CM) is becoming popular in describing these interventions; however, a consensus on its definition has not yet been reached. There are no consensual curricular outlines and/or competencies, and the potential for addressing food security and equity has not yet been fully developed. We believe that consensual competencies will formalize CM and ensure appropriate outcomes followed by improved assessments of learners, thus promoting CM research and further implementation of this novel nutrition education approach.

## 1. Introduction

### 1.1. Improved Nutrition for the Health of the Planet and Human Kind

Most premature adult deaths and chronic diseases, with their associated costs, are directly related to unhealthy behaviors, particularly poor nutrition [1]. Public health morbidity and mortality, and the financial burdens accompanying a national population with high rates of obesity and associated chronic diseases, continue to rise despite strong scientific evidence supporting healthy nutrition as an effective means of both prevention and treatment [2].

The 2022 White House Conference on Hunger, Nutrition, and Health, the first of its kind in 50 years [3], addressed the diet-related, chronic disease epidemic and emphasized the importance of nutrition equity and security in preventing chronic diseases. Disparities have been repeatedly observed in areas with systemic hurdles to accessing education [4], healthcare, and affordable, healthy food [5,6]. These stark manifestations of longstanding health inequities that result in the overconsumption of energy-dense foods have emerged from a body of evidence linking obesity and chronic diseases with vulnerable populations [5].

Despite the progress in agriculture and food systems, diets are not sustainable [7], and climate change further challenges the ability to feed the world. The International Food Policy Research Institute estimates that there will be a 20% increase in the number of malnourished children globally by 2050, compared to the estimated number in the absence of climate change [8]. What and how we eat also impacts the environment, with 26% of greenhouse gas emissions globally attributed to the food supply chain [9]. In response to the urgent need to address the climate emergency, research assessing the environmental impacts of various dietary habits has increased and has found the benefits of plant-based home cooking [10,11,12]. This led to new recommendations for sustainable diets with low environmental impact, which will contribute to food and nutrition security and healthy lifestyles for present and future generations [13]. The food we eat today directly impacts our health and the availability and cost of food tomorrow [14].

### 1.2. The Need for an Improved Nutritional Education Paradigm

The knowledge of what constitutes a healthy diet does not necessarily elicit lasting change [15,16] and the formation of new habits [17]. This discrepancy between knowledge and long-term nutritional behavioral change can be attributed to numerous cultural, social, and personal barriers, including poor planning, unhealthy food environments, and underdeveloped culinary skills [18]. Food choice is multifaceted and shaped by various factors, including the structural environment, community influences, resource access, and human emotions and behavior [19,20,21,22]. In addition, a recent survey found that consumers continue to say taste is the most impactful factor on their food purchase decisions [23]. A new paradigm must focus on improved self-management [24] and culinary skills and consider cultural and psychosocial factors, as well as food access and sustainability. This may better educate the public to adopt healthy nutrition, thus improving their health and the health of the planet.

A novel food classification supports a focus on culinary skills. Whereas nutrition scientists have traditionally classified foods by macronutrients, the novel NOVA classification categorizes foods according to “the nature, extent and purpose of the industrial processing they undergo” [25]. According to NOVA, ultra-processed foods (UPFs) are industrial foods and drinks made of food-derived substances and additives, usually containing little to no whole foods [25]. Common UPFs include packaged snacks, reconstituted meat products, and pre-prepared frozen meals [25]. While the NOVA limitations include being too broad and leading to discordance with other nutrient-based recommendations [26], recent studies have demonstrated that UPF consumption is positively correlated with cancer [27], cardiometabolic disease [28], and overall mortality [29]. However, 57% of the calories consumed by adults in the United States are from UPFs [30], and this phenomenon is spreading globally, including countries with established and deeply rooted culinary culture [31]. Consumption of UPF calories amongst children is 67% [32], indicating that UPF consumption may continue rising. It was recently suggested that considering the causality between the increase in UPF in the food supply and the rise in chronic diseases, a reduction in UPF consumption is required [33]. A new nutritional education paradigm, focused on culinary skills, may also lead to a home cooking habit, thus decreasing the consumption of UPFs and improving nutrition and health.

### 1.3. The Emerging Field of Health-Related Culinary Education

Innovative culinary education interventions that aim to improve food preparation skills and eating patterns have recently emerged [34,35]. These interventions, often practiced in teaching kitchens, are delivered by physicians, [36] chefs [37], health coaches [38,39], and dietitians [40]. They have been found to improve cooking confidence [41] and nutrition [42]. However, their long-term impact on health and the environment is yet to be established [34,35]. Furthermore, while cultural sensitivity and food access are key elements in culinary preparation, the ability of these programs to address health disparity is still underdeveloped.

The terms Culinary Medicine (CM) and culinary nutrition are becoming popular in describing these health-related culinary interventions. The American College of Lifestyle Medicine (ACLM) defines CM as an evidence-based field that merges nutrition and culinary knowledge with the skills needed to assist patients in maintaining health and preventing and treating food-related diseases by choosing high-quality, healthy food in conjunction with appropriate medical care [43]. Although a few publications suggest definitions for CM [44,45,46], its definition varies according to its user. There are currently no agreed-upon definitions, learning objectives, competencies, or curricula outlines, nor is there a consensus on CM’s potential in addressing nutrition security and the planet’s health. Our goal is to reach a consensus about the definition of CM, including a curricular outline and patient competencies.

Culinary Medicine is inherently interdisciplinary, blending medicine, nutrition, culinary arts, behavioral change, and public health. Due to its emergence across various professions, efforts toward a shared goal must be unified. We are planning a working group, funded by the Harvard Radcliffe Institute, that will bring together an interdisciplinary team of leaders in public health, culinary arts, nutrition and dietetics, lifestyle medicine, health coaching, behavioral change, social equity, and environmental science. We want to create (1) a consensual definition of CM, including a curricular outline and competencies for patients, and (2) structured cultural and socioeconomic adaptations that will improve personal and planetary health, and (3) structural adaptation for professional training in CM. Once a blueprint describing the consensus is finalized, we plan to validate it through a modified Delphi process, and then an international generalization will be required.

## 2. Discussion

### 2.1. Patient Education

A collective effort of leaders mapped and compared the first CM programs in 2016, which demonstrated several gaps in CM programming [42], including (1) the lack of a dominant behavioral change component, (2) a focus on teaching kitchens (whose drawbacks include low accessibility and high running costs), and (3) the need for a consensual curricular outline [47]. During the past decade, the field has expanded, and major CM initiatives have been developed to disseminate the new field within and outside of the US, including countries with established and deeply rooted culinary culture. These include the Teaching Kitchen Collaborative [48], the ACLM Culinary Medicine Curriculum [39], and the CHEF Coaching program [49]. Several current CM programs include virtual offerings [38,39,49] and a focus on behavioral change principles [50], however, a consensual outline has yet to be achieved. While the lack of environmental aspects was not considered one of the CM gaps, emerging evidence connecting healthy nutrition and sustained environment requires urgent action [12]. Discussing both the health and environmental benefits of various foods and cooking techniques could be a two-pronged approach, motivating the adoption of new home cooking habits.

Major CM initiatives include education about cost-saving grocery shopping, minimizing food waste and cultural preferences. However, significant adjustments are needed to address the social determinants of health [51], including improved access to healthy food. Nutrition equity and security include being able to purchase food for income-challenged individuals [26]. We aspire to build a generic roadmap that will adjust the consensual outline to the needs of specific cultures and socioeconomic statuses, thus leveraging CM to improve nutrition equity and security along with planetary health.

Promoting CM requires rigorous research that evaluates patient outcomes. While there are some limitations to the association between food preparation and health outcomes [52], systematic reviews found that CM programs have a short-term positive impact on participants’ home cooking and nutrition [34,35]. The first multicenter randomized control trials evaluating a CM program demonstrated positive outcomes up to one year [53], and a few recent prospectives CM trials demonstrated a decrease in UPF consumption [54,55]. To continue expanding CM, the need for rigorous studies to evaluate the long-term impact of CM programs on patient nutrition and health was identified [34,35]. Patients’ consensual competencies are critical instruments in the evaluation of these programs, allowing rigorous research and meta-analyses from multiple centers, thus furthering CM implementation in clinical settings.

Although long-term outcomes from CM programs are yet to come, CM has already been integrated into US healthcare reimbursement models, indicating its promise. Physicians can incorporate CM into billable shared medical appointments [36]. Dietitians or Dietetic Technicians can utilize CM as part of group or individual medical nutrition therapy [56,57], and health coaches can apply CM strategies during sessions [58]. Major institutions such as Kaiser Permanente and the Veterans Health Administration have already implemented CM to their programs [59]. Additionally, the Centers for Medicare and Medicaid’s 1115 waivers and health-related social-need reimbursement can cover the cost of food to address health disparity [60,61]. CM training programs may be offered to physicians [62] so they can prescribe nutrition or refer patients to another professional, like registered dietitians, who can augment their nutrition knowledge. CM training programs may be also offered to certified health coaches who can effectively combine culinary education and coaching competencies that enable a sustainable mindset and behavioral change [63] and to chefs who already work in healthcare organizations as food service staff and can expand their role as educators [37]. Delivering CM could also be done through coordination with training in the community, such as the Expanded Food and Nutrition Education Program (EFNEP) [64], SNAP-Ed [65], and Cooking Matters [66]; and start as early as primary and secondary school education. While the professions that will deliver CM programs and the setting have yet to be defined, this interprofessional effort is also an opportunity for a united voice promoting the contribution of healthy nutrition to the health of the planet and its inhabitants.

### 2.2. Education of Health Professionals

Although several individual and systemic factors have been implicated in developing and reinforcing healthy behaviors, it remains critical that clinicians assist their patients in facilitating healthy behavioral changes [67]. Unfortunately, few clinicians feel comfortable doing this, largely due to a lack of training [68]. Therefore, an impetus for reforming health professional education to address diet and other lifestyle behaviors has been bolstered by several significant initiatives [69]. These include the Bipartisan Policy Center’s call to incorporate nutrition training in all phases of medical education [70], the New England Journal of Medicine perspective that implored medical schools to emphasize health and health promotion rather than merely disease and diagnosis [71], and the Lifestyle Medicine Education (LMEd) Collaborative that was founded to transform medical education by including lifestyle medicine curricula in US medical schools [72].

Over the past decade, the University of South Carolina’s lifestyle medicine curriculum, open-sourced on the LMEd website [73], has garnered interest from dozens of US and international medical schools. It has become the official undergraduate medical education strategic partner of ACLM. Over the past ten years, ACLM has empowered 95 medical schools to start lifestyle medicine interest groups and bring nutrition education into the curriculum [74]. While lifestyle medicine curricula are growing, most nutritional content is still related to biochemistry, vitamin pathways, and substrate utilization [75]. Presenting a novel curricular outline focused on practical food-related and culinary-related knowledge and skills can positively impact patient eating behaviors.

The CM curricula, which augment nutritional knowledge with culinary skills, have started to appear in medical education programs [76]. These include the CHEF Coaching program [63], Health Meets Food [77], and Healthy Kitchens, Healthy Lives [26], with thousands of graduates implementing CM globally [78,79]. There is an urgent need for a standardized, consensual CM curricular outline that also meets the American College of Graduate Medical Education (ACGME) guidance for improving education in nutrition [80] to allow for further expansion, including undergraduates and residency programs. Recently, a consensus for proposed nutrition competencies for medical education was published [81]. To better train physicians and other healthcare professionals in food and nutrition, there is a need for a consensual CM curriculum within the framework of pre-existing and novel lifestyle medicine and nutrition education programs.

## 3. Conclusions

CM is an innovative field that facilitates fertile discussion among seemingly unrelated disciplines—culinary arts, environmental science, nutrition and dietetics, behavioral change, public health, lifestyle medicine, health coaching, social equity, and medicine. Such discussion has the potential of establishing a consensual CM curricular outline and competencies. This will identify agreed-upon evaluation tools, thus promoting CM research and leading to further implementation of this novel nutrition education approach. In addition, recommendations for formal cultural, social, and global adjustments can position CM as a viable solution to nutrition equity and security issues, thus improving the health of our nation and the planet.

The time has come to initiate this multidisciplinary discussion of two urgent, major worldwide challenges: the climate emergency and the rise in chronic disease. To do this effectively, we must now define the field of CM. We must create a consensual CM curriculum outline, as well as competencies for patients that can be used to enhance clinical care and health professional education. This will lead to a better understanding of the format of the team that will provide the education and the recommended setting. We believe that standardizing CM will play a key role in promoting practical, accessible, scalable, equitable, and evidence-based CM programs that have the potential to benefit patient and population health, increase nutrition equity and security, and improve the health of our planet.

## Data Availability

No new data were created or analyzed in this study. Data sharing is not applicable to this article.

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
