# Peer review of "Defining Culinary Medicine: A Call for Consensus on Competencies to Improve Nutrition"

_nutrients, 2025, doi:10.3390/nu17091403_

Round 1

Reviewer 1 Report

Comments and Suggestions for Authors

There  

what is the evidence that CM would lead to increased home coming?

The evidence for the effect of UPF is primarily observational.  That isnt a great argument for your call to action.    You need a food scientist on your team.

A positive of this paper is information that is not known throughout medical education   (lines 154 -167) or have not been successful when tried.  More detail would be valuable and further your argument

I may have overlooked it, but did you cite the Nutrition Summit report, from ACGME?  would be good to mention its conclusions and how that has helped or not to move nutrition education forward.  What is AAMC doing?

Reviewer 2 Report

Comments and Suggestions for Authors

The presented announcement presents an interesting and at the same time very sad topic, namely the loss of the ability to prepare meals independently in households and the resulting health consequences associated with excessive consumption of highly processed food. However, I believe that both the form of the presented announcement and the scope of issues it touches on do not qualify it for publication in the journal "Nutrients". The authors indicate the need for changes in the education system, but mainly refer to the American education system. It seems, for example, that this problem does not concern or concerns to a lesser extent societies with an established and deeply rooted culinary culture, e.g. European (France, Spain, Italy) or Asian (Chinese or Indian cuisine). In turn, issues related to malnutrition and lack of access to food in societies with a low material status do not result from a lack of knowledge or education but from low food availability and general poverty. For this reason, I believe that this undoubtedly important manifesto should be published in a journal dedicated strictly to American society, e.g. American journal of public health or American journal of education etc.

By the way, please note two editorial issues:

1) missing keywords

2) line 173 missing brackets and space at ref. no. 56

Round 2

Reviewer 1 Report

Comments and Suggestions for Authors

THE EDIT OF THE SENTENCE UNDER POINT 3, IS NOT PARTICULARLY HELPFUL.. PERHAPS YOU COULD GIVE EXAMPLE OF WHERE ,, WHAT AND HOW SUCCESSES IN GETTIN INTRO REIMBURSEMENT MODELS.. IS HAPPENING.. OR GIVE REFERENCES THAT COULD BE FOLLOWED TO GET THE HOW-TO INFORMATION OR EXAMPLES

Reviewer 2 Report

Comments and Suggestions for Authors

I appreciate the corrections made.

Author Response

No edits were suggested